# Crimean–Congo Hemorrhagic Fever Virus: Progress in Vaccine Development

**DOI:** 10.3390/diagnostics13162708

**Published:** 2023-08-19

**Authors:** Aykut Ozdarendeli

**Affiliations:** 1Department of Microbiology, Faculty of Medicine, Erciyes University, 38039 Kayseri, Türkiye; aozdarendeli@erciyes.edu.tr; 2Vaccine Research, Development and Application Centre (ERAGEM), Erciyes University, 38039 Kayseri, Türkiye

**Keywords:** Crimean–Congo hemorrhagic fever virus, hemorrhagic fever, immune response, animal models, vaccine development

## Abstract

Crimean–Congo hemorrhagic fever virus (CCHFV), a member of the *Nairoviridae* family and *Bunyavirales* order, is transmitted to humans via tick bites or contact with the blood of infected animals. It can cause severe symptoms, including hemorrhagic fever, with a mortality rate between 5 to 30%. CCHFV is classified as a high-priority pathogen by the World Health Organization (WHO) due to its high fatality rate and the absence of effective medical countermeasures. CCHFV is endemic in several regions across the world, including Africa, Europe, the Middle East, and Asia, and has the potential for global spread. The emergence of the disease in new areas, as well as the presence of the tick vector in countries without reported cases, emphasizes the need for preventive measures to be taken. In the past, the lack of a suitable animal model susceptible to CCHFV infection has been a major obstacle in the development of vaccines and treatments. However, recent advances in biotechnology and the availability of suitable animal models have significantly expedited the development of vaccines against CCHF. These advancements have not only contributed to an enhanced understanding of the pathogenesis of CCHF but have also facilitated the evaluation of potential vaccine candidates. This review outlines the immune response to CCHFV and animal models utilized for the study of CCHFV and highlights the progress made in CCHFV vaccine studies. Despite remarkable advancements in vaccine development for CCHFV, it remains crucial to prioritize continued research, collaboration, and investment in this field.

## 1. Introduction

Crimean–Congo hemorrhagic fever virus (CCHFV) is a member of the genus *Orthonairovirus* in the family *Nairoviridae* and the order *Bunyavirales* [1,2]. The viral genome is composed of three negative-sense RNA segments: small (S), medium (M), and large (L). The S segment encodes the nucleoprotein (NP), while the M segment encodes the glycoprotein precursor (GPC), which later forms mature Gn, Gc, and several nonstructural proteins such as mucin, GP38, and NSm [3,4,5,6]. The L segment encodes the L protein responsible for viral RNA synthesis, which includes the RNA-dependent RNA polymerase (RdRp) and an ovarian tumor (OTU) protease domain (for which the L protein is crucial for viral RNA synthesis) and includes the RNA-dependent RNA polymerase (RdRp) and an OTU protease domain that may aid in evading the host’s innate immunity (Figure 1a,b) [7,8,9].

It was initially detected in Soviet soldiers in Crimea during the 1940s. In the 1960s, a virus with similar symptoms to the Crimean virus was discovered in the Belgian Congo (currently known as the Democratic Republic of the Congo). Further studies revealed that both viruses were antigenically identical, leading to the virus being named CCHFV [10,11,12,13]. CCHFV circulates within an enzootic cycle that involves ticks and vertebrates [14,15,16]. Although CCHFV has been isolated in multiple tick species, *Hyolamma* ticks serve as the host and biological vector for CCHFV due to its extensive geographical range, which closely correlates with the distribution of CCHF cases [17,18]. Ticks can transmit CCHFV vertically from one generation to the next, transovarially from one developmental stage to another, sexually from males to females during copulation, or through cofeeding from one tick to other ticks feeding on the same non-viremic host [19,20,21,22]. CCHFV infects a wide range of both wild and domestic animals. However, infections in these animals are typically asymptomatic, but they exhibit viremia for more than five days, which helps the maintenance of CCHFV in nature [14,17,18,19,20,21,22].

CCHFV is found in a vast geographic region, from western China to Africa, the Middle East, Spain, and the Balkans [23,24,25,26,27,28,29,30,31,32,33,34,35,36,37,38,39,40]. Turkey has been experiencing CCHF epidemics since 2002, with the number of cases increasing significantly in recent years [41,42,43,44,45]. Similarly, some Balkan countries have reported regular cases of the disease [46]. CCHF was first reported in Spain in 2016, although tick surveillance studies had already shown the presence of CCHFV in the area [36]. This suggests a shift in the main vector’s geographic distribution, as there were no previous reports of autochthonous human cases in Spain [47,48,49]. Ticks primarily disperse over long distances only through their hosts [50]. Therefore, changes in tick populations are primarily associated with bird migrations or expansions of host populations. The geographical spread of tick populations is concerning as infected ticks transported to non-endemic areas can spread the disease to humans [48,51,52,53]. Furthermore, uninfected ticks introduced to a new area may establish populations that can sustain the virus after its introduction. The expansion of CCHF’s geographic distribution is also driven by several factors, including global warming, the increasing of human mobility, and human activities such as deforestation and agricultural growth, leading to more people coming into contact with infected ticks and animals [54,55,56,57]. Transporting livestock is a widespread practice across the world, and its contribution to the spread of CCHF cannot be overlooked. The movement of animals across borders or the transportation of infected ticks from endemic regions can initiate new CCHF outbreaks in non-endemic regions [58,59,60,61,62].

Human beings are regarded as accidental hosts of the CCHFV. Humans become infected through tick bites or exposure to crushed infected ticks during agricultural activities. Another significant source of infection is the blood of infected agricultural animals, which can be viremic but not display any symptoms of disease [14,17]. Nosocomial transmission contributes to the spread of CCHFV, leading to higher fatality rates compared to those resulting from tick bites. Several cases of nosocomial outbreaks have been linked to infected blood or needle-stick injuries during patient care [63,64,65,66,67]. CCHFV infection in humans can cause mild to severe symptoms, including high fever, malaise, myalgia, and gastrointestinal distress, typically after a short incubation period of about a week. Severe cases can result in hemorrhagic disease with a fatality rate ranging from 5 to 30%, often due to disseminated intravascular coagulopathy, shock, and/or multi-organ failure [63,64,65,67]. Due to its high fatality rate, widespread vector, and the absence of effective medical countermeasures for prevention and treatment, CCHFV is classified as a high-priority pathogen by the World Health Organization (WHO) [68].

## 2. Immune Response to CCHF Infection

Effective control of CCHFV infection in the host relies on immune responses from both the innate and adaptive systems. The innate immune system acts as the first line of defense against viruses by limiting viral entry, translation, replication, and assembly. Additionally, it facilitates the identification and elimination of infected cells, as well as the development of adaptive immunity through coordination and acceleration. Type 1 interferons (IFN-α/β) are produced by the host’s innate immune response against viruses. These responses are rapid and efficient, and they can be generated and secreted by all mammalian cells. These immunomodulators facilitate the expression of antiviral proteins, inhibit cell proliferation, and help regulate apoptosis [69,70,71]. 

The innate immune response is stimulated by CCHFV, leading to the production of IFNs and interferon-stimulated genes (ISGs). Andersson et al. conducted in vitro studies that confirmed the antiviral effect of IFN on CCHFV. The studies highlighted a substantial reduction of vRNA levels in cells treated with IFN, emphasizing the crucial role of IFN in controlling CCHFV replication [72]. Hawman et al. recently created a novel model using type I interferon-deficient mice, whereby infection with the human clinical isolate strain Hoti resulted in progressive illness characterized by several days of overt clinical signs [73]. This model also demonstrated the induction and release of IFNs, the subsequent upregulation of ISGs, and the involvement of the host’s innate immune response to CCHFV [74]. Bente et al. confirmed the crucial role of IFN in combating CCHFV using a STAT-1 KO mouse model, where STAT1 is a central component of IFN signaling pathways [75]. In 2012, CCHFV-infected IFN1-deficient mice (IFNAR^−/−^) exhibited clinical symptoms resembling CCHF, whereas wild-type mice remained asymptomatic, highlighting the importance of IFN1 in preventing CCHFV infection [76]. IFN is also crucial in controlling infections and preventing diseases in animals, even those with compromised adaptive immune systems.

Lindquist et al. demonstrated the temporary suppression of the immune response in various mouse strains (including wild-type and those with impaired adaptive immunity such as NOD/SCID, Prf1^−/−^ and Rag2^−/−^) by utilizing an anti-IFNAR1 monoclonal antibody (mAb) [77]. IFN responses play a significant role in determining disease severity. Studies indicated that polymorphisms in toll-like receptor genes (TLR7, 8, 9, and 10) are associated with increased illness severity in Turkish CCHF patients, thereby emphasizing the crucial role of TLRs as an immune-sensing pathway in controlling the virus [78,79]. Despite IFNs being crucial in the host’s immune response, CCHFV utilizes various strategies to evade and counteract the innate immune response. These include removing the 5’ triphosphate group from the viral genome to avoid RIG-I recognition, delaying IRF-3 activation through the particle recognition pathway, and downregulating NF-kappaB activation [80,81,82]. Additionally, studies have shown that CCHFV can suppress the body’s innate immune system by encoding an ovarian tumor-related deubiquitinase (OTU) domain. This domain deubiquitinates proteins involved in the body’s signaling pathways, thereby inhibiting innate immune responses including the antiviral response mediated by ISG15 modifications. In addition, the OTU domain has both de-ISGylation and deubiquitinase activity, which are important for viral pathogenesis [8,83].

Apoptosis can serve as a significant innate response to viral infections. It is a critical process involved in viral infections, as its suppression or induction determines the level of infection spread. Viruses can either inhibit host cell apoptosis, which is a defense mechanism against infections, to ensure their survival or on the contrary promote apoptosis to eliminate uninfected immune cells and facilitate viral spread [84,85]. As with many other viruses, CCHFV also has the ability to regulate apoptosis. CCHF infection induces TNF-α and FasL-mediated apoptosis in cell culture [86]. CCHFV NP inhibits caspase 3 and caspase 9 activation, prevents apoptosis initiated by BAX, and curtails the release of cytochrome c from mitochondria [87]. However, the specific point in the intrinsic apoptosis pathway where NP disrupts activation is yet to be determined. Despite the CCHFV NP inhibiting activation in the intrinsic pathway of apoptosis, the CCHFV NSs, a cryptic ambisense product of the NP, has been found to disrupt mitochondrial membrane potential, inducing apoptosis by activating caspase 3/7 and cleaving poly ADP-ribose polymerase. Furthermore, the presence of a conserved DEVD motif in the virus’ NP that can be cleaved by host caspase 3 implies a regulatory role in the virus’s life cycle [88]. 

Pro-inflammatory cytokines and chemokines, which are produced by various immune cells in response to viral infections, play a crucial role in the innate immune response against viral pathogens [89]. When a virus enters the body, immune cells such as macrophages and dendritic cells recognize it through pattern recognition receptors (PRRs) on their surface. These PRRs identify pathogen-associated molecular patterns (PAMPs) that are unique to the invading virus. Upon recognition of viral PAMPs, immune cells release pro-inflammatory cytokines [89,90]. While pro-inflammatory cytokines and chemokines are critical for the innate immune response to viral infections, excessive or uncontrolled release of these molecules can lead to tissue damage and inflammatory disease [91,92]. CCHFV initially targets immune cells such as dendritic cells, macrophages, and monocytes [93]. This results in the production of pro-inflammatory cytokines, including TNF-α, IL1, IL6, IL8, IL-12, IFN-γ, MCP-1, and MIP-1b [94]. Pro-inflammatory responses in severe or fatal diseases can lead to vascular dysfunction, disseminated intravascular coagulation (DIC), organ failure, and shock [95]. Increased levels of TNF-α, IL-8, IL-9, IL-15, IP-10, and MCP-1 are associated with disease severity and negative outcomes in patients from Turkey, Albania, and Kosovo [17,94,95,96,97]. Researchers have also found that the secretion of sTREM-1 by myeloid cells enhances inflammatory responses during CCHF virus infection, though it remains to be empirically demonstrated how excessive levels of these inflammatory agents may drive pathogenic processes [98,99].

While innate immunity serves as the initial protective response against CCHFV infection, a potent adaptive immune response is also essential for effective control of the infection. Anti-CCHFV IgM and IgG antibodies are detectable within 7–9 days from symptom onset [100,101,102]. IgG antibodies typically peak during the second to third week and can remain for up to 3 years. In contrast, IgM titers decline within 3 weeks and become undetectable between 3–5 months after disease onset. The absence of serum antibodies has been linked to higher mortality rates in CCHF patients, suggesting that antibodies may provide protection against fatal CCHFV infection [97,100]. Kaya et al. assessed serial antibody responses on 31 patients with CCHF, 11 of which were fatal cases. The study revealed that all surviving patients had a positive IgG titer within 9 days of onset, whereas none of the fatal cases showed such a response at the same timepoint [103]. In a study conducted on 24 patients with 43 samples, it was observed that quantitative IgG levels and viral loads had a correlation, and none of the fetal patients developed positive IgG titers. Only one sample from nine survivors taken less than nine days after the onset of the disease showed positive IgG titers. There was no correlation between death or viral load and IgM positivity [104]. In a study involving 46 confirmed cases of acute CCHFV infection in Kosovo, it was discovered that there was no correlation between the presence of IgM antibodies and clinical classification. Furthermore, only 5 out of the 34 patients who survived the disease exhibited IgG antibodies [95]. 

Neutralizing antibodies (NAbs) usually appear by day 10 of the illness. They are usually found at low levels in CCHF survivors, but are undetectable in fatal cases of CCHF [97,100]. This suggests that antibodies may play a crucial role in protecting individuals from lethal CCHFV infections. As of now, the study has revealed that mAbs and neutralizing mAbs specific to CCHFV have exclusively been derived from mice [105]. Among the isolated antibodies, three Gc-specific NAbs have demonstrated the ability to neutralize multiple strains of the virus [105,106]. However, despite their cross-neutralizing activity, these NAbs have not been effective in providing protection against CCHF in experiments conducted on mouse models. These epitopes are not associated with the production of NAbs that contribute to the immune response against CCHF. Three Gc-specific NAbs cross-neutralize various strains but are ineffective in protecting mouse models of CCHF [105,106]. Studies have shown that although mAbs targeting pre-Gn and/or GP38 lack neutralizing activity, they can still provide pre-exposure protection in mice [105,106,107,108]. These findings suggest that non-NAbs may also have the ability to protect against CCHF through other mechanisms besides neutralization, as evidenced by their ability to confer protection in fatal CCHF challenges. The GP38-targeting mAbs effectiveness in providing protection depends on complement activity. This finding suggests that the antibody’s effector functions, such as complement-mediated lysis and phagocytosis, play a crucial role in protecting against CCHFV [107]. However, a recent study illustrated the efficacy of bispecific antibodies (bsAbs) by incorporating variable domains from wide NAbs to boost their antiviral efficacy. The study found one bsAb to be particularly effective as it provided therapeutic protection against CCHFV with a single dose [109]. Thus, it is currently unclear whether there is any relationship between the neutralization antibody responses and positive disease outcomes. 

Several studies have found a strong relationship between a high viral load and fatality rate, with some identifying a viral load of ≥10^8^ copies/mL as a significant predictor of fatality [63,95,104,110]. While a reduction in CCHF viral load has been connected with the presence of antibodies in clinical infections, the production of antibodies is not always linked to the clearance of the virus. During the first week of infection, viral loads generally decrease irrespective of IgG levels, indicating the critical role of cellular immunity [95,104]. A study conducted with STAT1-deficient mice showed that CD4+ and CD8+ T cells were activated early against CCHFV infection [75]. Studies on infected mice have demonstrated that T cells play a critical role in controlling CCHFV infection. They limit the virus’s spread and prevent further infection by producing antiviral cytokines upon rapid activation [73,74]. 

The maintenance of CCHFV-specific T cells for an extended period after infection suggests that memory T cells may offer long-term immunity against the virus, responding rapidly to future exposure and serving as a lasting defense [111]. The study on a DNA vaccine highlights the significance of a TH1 response for effective protection [112]. It was shown that CD8+ T cell responses in human survivors lasted for 13 years after acute infection. Most T cell epitopes were found on the NP, but there were two instances of reactivity to GC-derived peptides. None of the epitopes were considered immunodominant [113]. Lindquist et al. found that an IFN blockade antibody treatment in mice effectively controlled CCHFV through adaptive immune responses, specifically cytolytic T cell activity, while avoiding liver damage, which is a common issue caused directly by CCHFV [77]. The removal of CD4+ or CD8+ T cells in mice infected with the virus resulted in a higher mortality rate, highlighting the indispensability of these cells in survival [74]. Hawman et al. also found that the absence of CD4 T cells eliminated the host’s IFN-γ response and blocking IFN-γ signaling led to lethality in IFNAR^−/−^ mice, suggesting that cellular immunity and type II IFN may control the CCHFV infection. Despite the existing research on the role of the adaptive immune response in CCHFV infection, further studies are needed to determine the immune responses and their effector functions essential for protection.

## 3. Animal Models for CCHF

CCHFV infections are asymptomatic in animals despite infecting many species and causing detectable viremia. With the exception of humans, only newborn mice and rats were susceptible to CCHFV among vertebrates. Disease signs and lethality are observed in human and newborn mice and rats through intracranial or intraperitoneal inoculation, making newborn rodents the first animal models for CCHF research [100,114]. However, they could not be used as models owing to their immature immune systems. Adult mice genetically deficient in type I IFN due to IFNα receptor or signal transducer and activator of transcription 1 (Stat1^−/−)^ have been used as lethal models of CCHF disease [75,76]. Interferon-deficient mice typically develop a rapid onset of severe illness resulting in death within four days post-inoculation. These mice exhibit elevated levels of inflammatory cytokines, liver enzymes, liver lesions, and spleen lymphocyte depletion, which is comparable to the symptoms observed in humans infected with CCHFV. However, immunocompromised mice may not be suitable for vaccine testing or fully understanding immune protection against CCHFV, as IFN-I signaling is critical in generating mature antigen-presenting cells, T and B cell responses, and memory T and B cell pools [75,76]. 

Recently, a novel murine system that utilizes the MAR1-5A3 antibody targeting IFN-I receptor A has been created [77]. This method has been previously used to generate severe disease models for different viruses and has proved to be effective in inducing lethal or severe CCHFV infection in mice by producing a temporary blockade of IFN-I [115,116]. The MAR1-5A3 antibody induces a temporary IFN-I blockade in mice, leading to a predictable and severe CCHFV infection [77,117,118]. A major benefit of this model is that it can mimic the same phenotype as an IFN-I receptor knockout animal in almost any wild-type or transgenic mouse, without requiring cross-breeding. This system enables the vaccination of an animal with an intact immune system prior to virus exposure, reducing the potential impact of IFN-I disruption on secondary immune responses upon challenge. The disease resulting from the antibody-mediated IFN-I blockade model is almost identical to the disease observed in genetic knockout animals, with both displaying equivalent mean times to death. 

It was recently reported that a humanized mouse model has been developed for CCHFV [119]. They are excellent small-animal models that have been transplanted with human cells or tissues, additionally equipped with human transgenes. These animal models are particularly useful in analyzing human hematopoiesis and studying pathogens with a special affinity towards humans, especially those that have been conditioned to support the engraftment of human immune cells [120,121]. Spengler and colleagues created a new type of humanized mouse model by introducing human CD34+ stem cells into NOD-SCID-gamma Hu-NSG-SGM3 mice, which have severe immunodeficiency and impaired cytokine signaling resulting from the absence of mature T cells, B cells, and natural killer cells as well as deficiency in the common gamma chain. While exposure to CCHFV strains from Oman and Turkey showed diverse disease patterns, it was only the Turkish strain that resulted in fatal outcomes. The humanized mice, Hu-NSGTM-SGM3, exhibited severe histopathological changes in the brain and are a promising model for investigating the cellular and molecular factors contributing to CCHF-related neurological disease [119].

A mouse-adapted strain of CCHFV is capable of infecting immunocompetent wild-type lab mice, causing significant pathology in the liver and spleen, high viral loads in multiple tissues, and inducing inflammatory cytokine production. Upon convalescence, robust humoral and cellular immunity was observed. Five coding mutations were detected in this virus through sequencing: two were found in the viral NP along with one mutation that also affected the viral NSs, one in the NSm, and two in the L protein (111). Although sex-linked differences have occasionally been reported for the CCHF patients [122,123,124], female mice showed greater resistance to severe disease than male mice, which exhibited a sex-linked bias in disease severity. The severe disease observed in male mice shared similar characteristics with poor outcomes in human CCHF cases, such as increased production of inflammatory cytokines, prolonged viraemia, and greater tissue pathology [111]. 

Previous studies have shown that the infection of African green monkeys, baboons, and patas monkeys with CCHFV was not successful [125,126,127]. However, Haddock et al. have developed a cynomolgus macaque model of CCHF that was infected with a human clinical isolate of CCHFV (the Hoti strain), administering a high dose (5 log10 TCID50) through intravenous (IV) or combined IV and subcutaneous (SC) exposure. The infected animals showed viremia and developed a severe, occasionally lethal disease, with symptoms such as inflammatory immune responses, heightened liver enzymes, extended clotting times, thrombocytopenia, leukopenia, and fever. Four out of eight animals were humanely euthanized by day seven post-infection for meeting humane endpoint criteria. Animals that experienced severe disease had liver pathology, inflammatory cytokines, high viral loads, and coagulation disorders, which are similar hallmarks found in severe human disease [128]. In a study assessing the antiviral drug favipiravir’s effectiveness for CCHFV-infected macaques, only one of eight animals in the placebo group met euthanasia criteria by day eight, while the other animals had moderate disease outcomes [129]. Although this model has already been utilized to evaluate antivirals and vaccines, the disease outcomes in the model were variable.

Through the use of the cynomolgus macaque model, Cross et al. enhanced the understanding of CCHFV pathogenesis by comparing two distinct strains, Afg09-2990 and Hoti [130]. Two separate groups of cynomolgus macaques were intravenously injected with each strain, and their disease courses were closely monitored. All animals exhibited clinical illness, viremia, significant changes in clinical chemistry, hematology values, and serum cytokine profiles consistent with CCHF disease in humans. However, in contrast to the earlier study, all NHPs recovered fully, and none of the animals met the euthanasia criteria [130]. In a separate study conducted by Smith et al., it is worth mentioning that the persistence of CCHFV has been observed in the testes and latent tuberculosis granulomas of macaques infected with the virus. This observation suggests that the virus may have the ability to persist in immune-privileged sites [131]. Although the factors that contribute to disease variability are not yet fully understood, the use of cynomolgus macaques as a model has proven useful in preclinical evaluation of anti-CCHFV therapeutics and vaccines. 

## 4. Vaccine Targets of CCHF 

### 4.1. The Nucleoprotein (NP)

The NP is an essential component of CCHFV, as it plays a crucial role in the viral life cycle. The primary function of the NP is to encapsidate the viral RNA and form the ribonucleoprotein (RNP) complex (Figure 1a). The RNP complex is the basic unit of the virus and is required for viral replication and transcription. CCHFV NPs are involved in a range of important functions, many of which relate to interactions with components of the host cell. These interactions may facilitate the transport of viral RNPs, the evasion of the host’s immune system, and the regulation of apoptosis [3,132]. The NP is a promising vaccine target for CCHF due to its abundance and high immunogenicity [133]. The NP contains both B and T cell epitopes, making it capable of stimulating both the humoral and cellular immune responses [113]. The genetic diversity and multiple distinct lineages of CCHFV existing in different global regions are crucial factors to consider for the development of vaccines. The NP of CCHFV exhibits a remarkable degree of conservation across various strains of the virus, suggesting NP could provide protection against multiple strains of the virus [134,135].

### 4.2. The Glycoproteins (GPC)

The CCHFV M segment encodes a polyprotein that undergoes post-translational processing to form intermediate glycoprotein precursors, PreGn (140 kDa) and PreGc (85 kDa) (Figure 1b). These intermediate glycoproteins are subsequently processed to yield envelope glycoproteins (Gn and Gc), nonstructural M protein (NSM), as well as secreted non-structural proteins (GP160, GP85, and GP38) and mucin-like domain (MLD) [5,6,106,136] (Figure 1b). Gn and Gc glycoproteins are essential for virus attachment and envelope fusion with host cells [137,138]. One of the most promising approaches to developing a CCHFV vaccine is the use of glycoproteins Gn and Gc as antigens. These glycoproteins are located on the surface of the virus and are responsible for facilitating viral entry into host cells (Figure 1a). They are also the primary targets of the immune response during CCHFV infection, making them ideal candidates for vaccine development. Early studies indicate that antibodies that target Gc glycoprotein in the CCHF virus have neutralizing capabilities, while no such neutralizing effect has been observed for Gn-targeting antibodies [106]. It has been observed that neonatal mice can be protected from lethal CCHFV infection by both neutralizing and non-neutralizing mAbs antibodies, but these findings are limited due to the fact that neonatal mice do not accurately represent CCHF disease, thereby making it difficult to interpret the results. However, a recent study examined the effectiveness of murine mAbs in protecting adult mice from CCHFV infection. The findings showed that non-NAbs targeting the GP38 protein provided protection against lethal CCHFV infection in mature animals [108]. The study highlights the potential of developing antibody-based CCHFV countermeasures. 

The diversity of the M segment of CCHFV, particularly the region encoding the nonstructural proteins, has long been suspected to impact cross-reactivity and ultimately the neutralization ability against different strains [69]. Understanding this diversity is crucial for developing vaccines, antiviral therapies, and understanding the virus’s evolution and pathogenesis. Further studies will be needed to fully elucidate the mechanisms underlying this diversity and its impact on viral pathogenesis.

## 5. Platforms for CCHF Vaccine Candidates 

Animal models for CCHFV have historically been limited by the lack of suitable animal hosts. Prior to the discovery of CCHFV animal models, there were limited attempts to develop a vaccine for CCHF, and efficacy studies were not possible. However, in recent years, researchers have developed new animal models that closely mimic the disease in humans, providing a more accurate representation of the virus. Also, recent advances in biochemical and molecular techniques have enabled researchers to employ different vaccine platforms for developing CCHFV vaccine candidates (Figure 1c and Table 1). Below are summarized the various approaches and platforms used in vaccine development.

### 5.1. Inactivated Vaccines

In the 1960s, Soviet scientists developed an experimental CCHF vaccine using brain tissue from infected newborn laboratory mice and rats. The vaccine was developed by cultivating CCHFV in suckling mouse brain and subsequently inactivating the virus through chloroform treatment and heat exposure at 58 °C. The inactivated virus was subsequently absorbed onto aluminum hydroxide (Al(OH)_3_) [139]. It was approved and licensed in 1974 in Bulgaria and has been used in military and medical personnel and people living in endemic regions. To date, the only tested CCHFV vaccine for humans is the suckling mouse brain-derived vaccine, utilized exclusively in Bulgaria. Between 1953–1996, the Bulgarian Ministry of Health observed a reduction in CCHF cases from 1105 to 279 [140]; however, vaccination may not solely account for this reduction as factors such as changes to ecology and epidemiology may have also contributed. Data on the immunogenicity of the mouse brain-derived vaccine are limited. In 2012, a study showed that repeated vaccinations among healthy volunteers were associated with high levels of CCHFV antibodies, anti-CCHFV specific T cell activity, and low levels of neutralization activity (Table 1) [141]. However, this vaccine is only authorized for use in Bulgaria and not in other countries with at-risk populations due to potential allergic and autoimmune reactions, and it is unlikely to receive international approval due to safety issues and scalability limitations.

Our group developed a purified and formalin-inactivated CCHF vaccine candidate derived from cell culture in 2015. The inactivated vaccine was prepared by growing the CCHFV Turkey-Kelkit06 strain in cell culture, harvesting it, and inactivating it with formaldehyde. Alum adjuvant was added and administered through three inoculations of 5, 20, and 40 μg dosages to IFNAR^−/−^. Two weeks after the last immunization, the mice were challenged with a high lethal dose (1000 PPFU) of the CCHFV Turkey-Kelkit06 strain to test the vaccine’s efficacy. Immunization with the cell culture-based CCHF vaccine at doses of 5, 20, and 40 μg provided partial protection ranging from 60% to 80%, with a significant delay in time to death (Table 1) [142]. Until recently, there were no data available on the efficacy of the mouse brain-derived vaccine. Our recent study using the immune-suppressed (IS) mouse model investigated the potential efficacy of the mouse brain-derived vaccine. The challenge studies showed that the mouse brain-derived vaccine provided complete protection, but the cell culture-based vaccine more effectively stimulated CCHFV-specific antibody and T cell responses (Table 1) [117]. In a recent study, Engin et al. investigated the IgG and neutralizing antibody titers over a duration of one year in BALB/c mice following vaccination with the cell culture-based and mouse brain-derived vaccines. Consistent with previous findings, the cell culture-based vaccine resulted in higher IgG and neutralizing antibody titers compared to the mouse brain-based vaccine at all measured time points [143].

### 5.2. Subunit Vaccines

Subunit vaccines are a safe and effective vaccine preparation strategy that rely on viral proteins to generate an immune response without eliciting antibodies against unrelated antigens or viral particles. The CCHFV envelope glycoproteins Gn and Gc were expressed using insect expression technology in Drosophila Schneider 2 (S2) cells. Adjuvanted Gn or Gc ectodomains were able to generate NAbs, but they did not provide protection to STAT-1 knockout mice after challenge (Table 1) [144]. Xia et al. used an affinity purification method based on a Gram-positive enhancer matrix-protein anchor (GEM-PA) surface display system to exhibit eGN, eGC, and NAb epitopes (NAb, aa1443, and 1566 of the M gene in IbAr10200 strain) of glycoprotein. The immunogenicity of these epitopes was evaluated in BALB/c mice [145], but the efficacy of the vaccine candidates was not assessed. In the study conducted in 2023, three vaccine candidates (rvAc-Gn, rvAc-Np, and rvAc-Gn-Np) expressing CCHFV’s glycoprotein Gn and NP on a baculovirus using the insect baculovirus vector expression system were evaluated for immunogenicity in mice. The results indicated limited immunogenicity for rvAc-Gn-NP, while rvAc-Gn elicited cellular and humoral immunity [146]. Once again, no vaccine effectiveness study has been conducted.

Plant-based vaccines offer scalable and cost-effective approaches for foreign gene expression in transgenic plants compared to conventional methods. In recent years, there has been considerable growth in producing human vaccine candidates in plants that address different targets [147]. Using plant cloning vectors, researchers introduced the Gn and Gc coding regions of CCHFV into transgenic tobacco plants for expression. Feeding the roots and leaves from these plants to mice resulted in oral/mucosal immunization that induced CCHFV-specific anti-Gn/Gc IgG and IgA antibodies in their serum and fecal material [148]. Nevertheless, the neutralizing ability of these antibodies was not evaluated, and there were no challenge studies conducted.

### 5.3. DNA Vaccines

DNA vaccines are a safe, efficient, and cost-effective means of inducing an immune response. Recombinant DNA is used to express antigens in antigen-presenting cells, inducing an immune response similar to viral infections. The delivered DNA can be translated into a desired protein in the cell cytoplasm, and the resulting peptide fragments can be presented by MHC class I and/or II molecules, promoting innate and adaptive immune responses. DNA vaccines have several advantages over other vaccines, including ease of manufacturing, improved safety, and simplified production. While there are some potential drawbacks to this approach, such as autoimmune responses and low immunogenicity, these can be addressed through innovative design [149,150]. Overall, DNA-based vaccines have significant potential for the development of vaccines against CCHFV. The first DNA vaccine development study against CCHFV dates back to 2006. A DNA-based vector was created to deliver the GPC of IbAr10200 CCHFV. Vaccination alone or combined with other DNA vaccines against Rift Valley fever virus, tick-borne encephalitis virus, and Hantaan virus showed NAbs detection in only 50% of vaccinated mice. Cell-mediated immune responses were not assessed and challenge studies were not conducted due to the unavailability of an appropriate animal model during that time [151].

A DNA vector encoding mature CCHFV envelope glycoproteins (Gn and Gc) as well as the NP of the IbAr10200 strain was intradermally immunized three times. IFNAR^−/−^ mice were successfully protected from lethal CCHFV challenge through the successful elicitation of both antibody and T cell immune responses [112]. In the same study, mice that received a VLP construct showed higher in vitro NAbs compared to the CCHFV DNA vaccine, although their protection was only partial. These results suggest that NAbs alone may not be sufficient to provide protection against CCHFV.

Garrison et al. evaluated the immunogenicity and protective efficacy of a DNA vaccine that expresses the M-segment glycoprotein precursor gene of CCHFV in two lethal mouse models of disease: IFNAR^−/−^ mice and a transiently immune suppressed (IS) mouse model [152]. In the study, the vaccine was administered via muscle electroporation at a dose of 25 μg, which stimulated a robust humoral immune response. After three vaccinations, neutralizing titers were detected in both mouse models. Both mouse models exhibited predominantly Th1 antibody responses, but the IS model had a significantly lower Th1/Th2 ratio, indicating a more balanced antibody response in immunocompetent mice. Although complete protection was not achieved in either mouse model, the survival rate was higher in the IFNAR^−/−^ model (71.4%) compared to the IS model (60%) (Table 1).

The same group conducted a recent study in 2021 to evaluate the effectiveness of two CCHFV DNA vaccines, namely CCHFV-M10200 and CCHFV-MAfg09, in mice [153]. The vaccine dose for CCHFV-M10200 was increased to 50 μg, administered three times, three weeks apart, resulting in 100% protection after a final vaccination and challenge with CCHFV-IbAr10200. Challenge studies also showed that CCHFV-MAfg09 provided complete protection against homologous CCHFV-Afg09-2990 challenge, while CCHFV-M10200 provided 80% protection against heterologous CCHFV-Afg09-2990 challenge (Table 1). They also found that a DNA vaccine expressing the GP38 region of CCHFV-IbAr10200 partially protected against homologous challenge, and high levels of anti-GP38 antibodies provided protection from CCHFV challenge, suggesting genetic diversity in the GP38 region is responsible for the diminished protection provided by the vaccines against heterologous challenge.

The study assessed the protection potential of a DNA vector expressing CCHFV’s NP and CD24 in mice, discovering that CD24 led to an induced immune response by regulating B and T cell proliferation [154]. CCHFV’s NP is a promising candidate for vaccination due to its ability to stimulate a balanced immune response. A study identified a complete protection in IFNAR^−/−^ mice with a DNA vaccine based on the nucleocapsid (Table 1). These results showed that the introduction of appropriate adjuvants for DNA vaccine immunization is a promising approach to enhance the immune response and efficacy of DNA vaccines.

Hu et al. recently developed three DNA vaccines that encode the NP, glycoprotein N-terminal (Gn), and C-terminal (Gc) of CCHFV [155]. These vaccines are fused with lysosome-associated membrane protein 1 (LAMP1) and have been tested for their immunogenicity and protective efficacy in a human MHC (HLA-A11/DR1) transgenic mouse model. Vaccination of mice with pVAX-LAMP1-CCHFV-NP was the most effective vaccine, inducing balanced Th1 and Th2 responses and providing effective protection against CCHFV transcription and tecVLPs infection. In contrast, pVAX-LAMP1-CCHFV-Gc elicited mainly specific anti-Gc and NAbs, while pVAX-LAMP1-CCHFV-Gn provided inadequate protection against CCHFV tecVLPs infection.

A DNA-based vaccine was tested in a cynomolgus macaque model for CCHFV [156]. The vaccine contains plasmid-expressed CCHFV strain Hoti NP and GPC, delivered through intramuscular injections with in vivo electroporation. The vaccine was well tolerated and induced CCHFV-specific antibody and T cell responses. However, the vaccinated macaques produced low levels of NAbs against CCHFV (Table 1). They also showed reduced viremia, clinical signs, and pathology following CCHFV challenge compared to unvaccinated controls. The DNA vaccine is the first to show efficacy in a non-human primate model of CCHF and supports the vaccine’s advancement into human clinical trials.

### 5.4. Virus-like Replicon Vaccines

Virus-like replicon vaccines (VRP) are virus-like particles that enter cells and undergo limited transcription and translation to synthesize proteins but do not produce infectious progeny. They are engineered virus genomes that express multiple proteins and lead to a strong immune response as they produce a high level of antigen expression in a single round of infection. The VRP vaccine lacks the M segment, which limits replication to one cycle and includes S and L genome segments from the IbAr10200 strain. However, co-transfection with a plasmid containing the optimized GPC of the Oman-98 strain enhances VRP generation and amplification for optimized cell entry. When tested in an IFNAR^−/−^ mouse model, the VRP vaccine, based on the IbAr10200 strain with the GPC sequence from the Oman-1998 strain, provides complete protection against lethal challenge following a single high dose (10^5^ TCID50 of VRP) subcutaneous vaccination. However, when a low dose (10^3^ TCID50 of VRP) of the vaccine was given, it was able to protect seven out of the nine mice (Table 1) [157]. This demonstrates that even a lower dose of the vaccine can still provide significant protection. Based on a related study, it has been discovered that the VRP candidate vaccine offers heterologous protection against CCHF disease. This protection was observed in IFNAR^−/−^ mice that were vaccinated with a single dose of VRP and subsequently challenged with CCHFV-Turkey and Oman-97 strains (Table 1) [158]. In a recent study, the efficacy of vaccinating IFNAR^−/−^ mice at different time intervals before exposure to the CCHFV was evaluated. It was observed that all non-vaccinated mice succumbed to the infection within 8 days, whereas mice vaccinated 14 or 7 days before the virus challenge were fully protected, while those vaccinated 3 days before showed symptoms but later recovered (Table 1). These findings suggest that the VRP vaccine could be used in shorter vaccination protocols to protect against severe disease outcomes [159].

Flavivirus replicons, which are viral self-replicating sub-genomic replicons, are a potent tool for studying viral genome replication, antiviral screening, and creating chimeric vaccines [160]. Recently, the Kunjin strain of West Nile Virus (WNV) was used to create flavivirus-based replicon virus-like particles for CCHF vaccine development. The C-prM-E genes in the WNVKUN replicon were replaced with the genes encoding the Gn and Gc glycoproteins of CCHFV to generate a replicon capable of expressing CCHFV proteins. The CCHFV Gn and Gc glycoproteins were expressed in the RVP vaccine platform; however, they induced a weak antibody response against them. T cell responses were not measured, and their protective efficacy in vivo was not assessed (Table 1) [161].

Alpha vector systems have emerged as a promising platform for developing prophylactic and therapeutic vaccines against infectious diseases [162]. A vaccine for CCHFV has been developed using the DNA-based Sindbis replicon platform. The vaccine, pSinCCHF-52S, replaces the structural proteins of the Sindbis virus with the NP gene of CCHFV, inducing NP-specific antibody and T cell responses with a Th1 skew [163]. However, the vaccine has not yet been tested for efficacy against CCHFV in a challenge model. The Venezuelan Equine Encephalitis Virus (VEEV) RNA replicon is another Alpha vector vaccine platform that has been used to produce potential CCHFV vaccines for NP (repNP), GPC (repGPC), and a combination of both (repNP + repGPC) [164]. The repNP vaccine elicited a strong antibody response but a weak T cell response, whereas the repGPC vaccine stimulated a weak antibody response but a strong T cell response. The repNP and repNP + repGPC vaccines provided complete protection after challenge with a heterologous strain of CCHFV in mice treated with an anti-IFNAR antibody blockade, whereas only 40% of mice immunized with repGPC were protected (Table 1).

### 5.5. mRNA Vaccines

The progress made in mRNA design, nucleic acid delivery technology, and the identification of new antigen targets has elevated mRNA vaccines to an exceptional tool for fighting emerging infectious diseases [165]. During the COVID-19 pandemic, mRNA-based vaccines have proven highly effective against the virus and are considered a robust alternative to traditional vaccines due to their potency, safety, and efficiency [165,166]. The study evaluated the efficacy of a conventional mRNA vaccine expressing NP from the non-optimized S segment of the Ank-2 strain of CCHFV. Single and booster doses were given, and challenge assays showed a 100% protection rate in the booster group and 50% in the single dose group, indicating lower effectiveness with a single dose (Table 1) [167].

In a recent study, Appelberg et al. designed two nucleoside-modified mRNA-lipid nanoparticle (LNP) vaccines that encoded either the CCHFV IbAr10200 glycoproteins (GcGn) or NP and tested them in both immunocompromised and immunocompetent mice. Challenge studies demonstrated complete protection for IFNAR^−/−^ mice vaccinated with either GnGc mRNA-LNP or NP mRNA-LNP (Table 1) [168]. The study does not provide conclusive evidence that a cellular immune response alone can protect against CCHFV, and it remains uncertain if antibodies are always necessary. However, the inclusion of NP in the vaccine can improve protection against different strains of CCHFV. Further studies are necessary to determine if cellular immunity through NP mRNA-LNP is enough to prevent CCHFV infection.

### 5.6. Viral Vector Vaccines

Recombinant viral vectors have been extensively studied as a promising vaccine platform due to their ability to express the antigens, stimulate antigen-specific immune responses, and generate potent antibody titers, all without the need for external adjuvants [169,170]. Multiple vaccine candidates have been developed for CCHF using various viral vectors, including the modified Vaccinia Ankara virus (MVA) [171,172], recombinant adenovirus type 5 (AdHu5) [173,174], recombinant chimpanzee adenovirus (ChAdOx2) [175], recombinant vesicular stomatitis virus (rVSV) [176], and recombinant bovine herpesvirus type 4 (BoHV-4) [174].

The MVA vector was used to create the CCHF MVA-GP vaccine candidate, and it encodes the entire M segment open reading frame (ORF) of the IbAr10200 CCHFV strain. Following intramuscular administration twice, the MVA-GP vaccine has shown the capability to induce NAbs and T cell responses leading to complete protection against intradermal lethal challenge in IFNAR^−/−^ mice (Table 1) [171]. Dowall et al. demonstrated that MVA-based vaccines effectively stimulate both immune system arms, essential in eliciting protective effects against lethal CCHFV challenge [177].

A recent study found that the ChAdOx2 CCHF vaccine, which employs a recombinant chimpanzee adenovirus to express the entire M segment of CCHFV, produced comparable results to a study using the MVA-GP vaccine. This study has been conducted to examine the immunogenicity and protection provided by the ChAdOx2 CCHF vaccine either alone or combined with the MVA-GP CCHF vaccine. They found strong antibody responses and IFN-γ-mediated cellular immunity upon administration of various vaccine combinations in both immunocompetent BALB/c and immune-deficient A129 mice. The study demonstrated that a single dose of the ChAdOx2 CCHF vaccine or homologous/heterologous prime-boost vaccination regimens resulted in full protection against CCHFV-induced disease in the A129 lethal mouse model (Table 1) [175].

The recombinant vesicular stomatitis virus vector vaccine (rVSV) for CCHF is another example that utilizes the GPC antigen. In a STAT-1/mouse model, a replication-competent recombinant rVSV that encoded the CCHFV GPC gene of the IbAr10200 CCHFV strain provided complete protection following a single intraperitoneal immunization. Conversely, a replication-deficient rVSV construct failed to provide protection against a lethal virus challenge administered intraperitoneally [176].

The MVA-NP vaccine candidate, which encodes the S segment ORF of the IbAr10200 CCHFV strain, failed to provide protection after a lethal challenge, even though it was capable of activating both arms of the immune system against CCHFV [172]. In contrast to the MVA-NP vaccine candidate, Zivcec et al. developed a promising NP-based candidate vaccine (Ad-N) that uses human adenovirus 5 to encode the NP of the CCHFV strain IbAr10200 [173]. The Ad-NP was able to provide partial protection in IFNAR^−/−^ mice against virus challenge. Additionally, a prime-boost strategy was used, which resulted in enhanced protection against the virus and reduced clinical symptoms compared to single-dose vaccination methods (Table 1). Another study confirms that using recombinant AdV-5 encoding NP from CCHFV can protect IFNα/β/γR- mice after CCHFV challenge (Table 1). Furthermore, the antibody passive transfer and T cell adoptive transfer experiments demonstrated a 50% survival rate of mice after a lethal CCHFV challenge [174].

BoHV-4 is another viral vector that has been used in the development of a vaccine for CCHFV. BoHV-4 possesses several characteristics that make it an ideal vaccine vector, such as its ability to stably express foreign genes, its replication capability in foreign hosts, and minimal pathogenicity in various hosts. When used in combination with a prime and boost strategy, the BoHV-4 vector encoding the full-length NP of CCHFV provided complete protection after lethal challenge (Table 1). Additionally, partial protection was observed in experiments involving antibody passive transfer and T cell adoptive transfer [174].

**Table 1 diagnostics-13-02708-t001:** The table provides a brief overview of various vaccine design platforms for Crimean–Congo hemorrhagic fever virus (CCHFV). Each platform utilizes different approaches to elicit an immune response against CCHFV antigens and potentially try to confer protection against the disease.

Vaccine Design Platforms	Strain Name and Types of Antigen	Animal Models	Doses and Vaccination Strategies	Specific Antibody Response	Neutralizing Antibody Response	T Cell Immune Response	CCHFV Challenge	Survival Rate %	References
Inactivated vaccines	Whole CCHFV from mouse brain	NE	Several thousand people took repeated vaccination.	NE	Yes	NE	NE	NE	[139]
	Bulgarian V42/81 strain; CCHFV whole antigen from mouse brain	NE	One group received a single dose, while the second group received four doses.	Yes	Yes	Yes	NE	NE	[140,141]
	Turkey-Kelkit06 strain; CCHFV whole antigen from cell culture (Vero-E6)	IFNAR^−/−^ mice and BALB/c mice with transiently immune suppressed by mAb-5A3	IFNAR^−/−^ mice: Administered (IP) at doses 5 μg, 20 μg, and 40 μg of inactivated vaccine on days 0, 21, and 42. BALB/c mice: Administered (IP) at doses 5 μg, 10 μg, and 20 μg of inactivated vaccine on days 0, 14, and 27.	Yes	Yes	Yes	Turkey-Kelkit06 strain; 1000FFU (IFNAR^−/−^) and 100FFU (Balb/C)	80% protected (IFNAR^−/−^), 100% protected (Balb/C)	[117,142,143]
Subunit vaccines	IbAr10200 strain; CCHFV Gn and Gc ectodomain	STAT1 knockout mice	Administered (IP) at doses 1.4 μg, 7.5 μg, and 15 μg of Gn and Gc on days 0 and 21.	NE	Yes	NE	IbAr10200 strain; 100PFU	Not protected	[144]
	IbAr10200 strain; extracellular region of Gn (eGn), extracellular region of Gc with truncation of C terminal (eGc), neutralizing antibody region of Gc (NAB)	BALB/c mice	Administered (SC) at doses 1 μg, 5 μg, and 20 μg of surface display protein of G-GP with eGn, eGc, at weeks 0, 3, 6, and 9.	Yes	Yes	Yes	NE	NE	[145]
	Chinese Xinjiang strain HANM18; Gn and NP from CCHFV expressed in baculovirus expression system as rvAc-Gn, rvAc-NP and rvAc-Gn-NP	BALB/c mice	10^7^ PFU, on days 0, 14, and 28.	Yes	NE	Yes	NE	NE	[146]
	CCHFV Iranian strain; Gn and Gc expressed in transgenic tobacco leaves	BALB/c mice	Feeding (leaves) at dose 10 μg of Gc/Gn at weeks 0, 1, 2, and 3. Feeding (roots) at dose 10 μg of Gc/Gn at weeks 0, 1, 2, 3.	Yes	NE	NE	NE	NE	[148]
DNA vaccines	IbAr10200 strain; GPC	BALB/c mice	Administered (gene gun) at dose 2.5 μg of the vaccine from each CCHFV + RVFV + HTNV + TBEV as total 10 μg, alone or combined at weeks 0, 4, and 8.	NE	Yes	NE	NE	NE	[151]
	IbAr10200 strain; Ubiquitin linked version of Gn, Gc, and NP	IFNAR^−/−^ mice	Administered (ID) at dose 15 μg of the vaccine three times with 4 weeks interval between first and second dose and 3 weeks interval between second and third dose.	Yes	Yes	Yes	IbAr10200 strain; 400FFU	100% protected	[112]
	IbAr10200 strain; GPC	IFNAR^−/−^ mice and C57BL/6 mice with transiently immune suppressed by mAb-5A3	Administered (IM) at dose 25 μg of GPC at weeks 0, 3, and 6.	Yes	Yes	NE	IbAr10200 strain; 100PFU	Protective efficacy 70% for IFNAR^−/−^ mice and 60% for transiently immune suppressed mice	[152]
	IbAr10200 strain; GPC from	C57BL/6 mice with transiently immune suppressed by mAb-5A3	Administered (IM) at dose 50 μg of GPC at weeks 0, 3, and 6.	Yes	NE	Yes	IbAr10200 and Afg09-2990 strains; 100PFU	100% protected against IbAr10200, 80% protected against Afg09-2990	[153]
	Ank-2 strain; NP with CD24	BALB/c for immunological responses and IFNAR^−/−^ mice for challenge studies	Administered (IM) at dose 50 μg of pV-N13 and 40 μg of pV-N13 with 10 μg of CD24 on days 0 and 14.	Yes	NE	Yes	Ank-2 strain; 1000 TCID_50_	100% protected	[154]
	IbAr10200 strain; NP, N terminal Gn, and C terminal Gc fused with LAMP1 to generate three candidate vaccines	Human MHC (HLA-A11/DR1)	Administered (IM) at dose 70 μg pVAX-LAMP1-NP, 70 μg pVAX-LAMP1-Gn, 70 μg pVAX-LAMP1-Gc, at weeks 0, 3, and 6.	Yes	Yes	Yes	IbAr10200 strain; 100 TCID_50_ CCHFV tecVLPS	Instead of measuring survival percentages, NanoLuc activities measured, NP had the lowest levels of NanoLuc activities in their liver, spleen, and kidney	[155]
	Hoti strain; Ubiquitin fused with GPC and NP	Cynomolgus macaque	Administered (IM) at dose 1 mg of pNP +1 mg of pGPC on days 0, 21, and 42.	Yes	Poor neutralization activity	Yes	Hoti strain; 1 × 10^5^ TCID_50_	Survival percentage was not assessed due to the non-uniform lethality of CCHFV in this animal model	[156]
Viral-like replicon particles (VRP) vaccines	Oman 98 strain; NP, GPC, and L segment	IFNAR^−/−^ mice	Administered (SC) at dose 10^5^ TCID_50_ or 10^3^ TCID_50_ of VRPs, single vaccination.	Yes	NE	NE	IbAr1020 strain; 100 TCID_50_	Low dose showed 77% protection and high dose showed 100% protection	[158]
	Oman 98 strain; NP, GPC, and L segment	IFNAR^−/−^ mice	Administered (SC) at dose 1 × 10^5^ TCID_50_ VRRs, single vaccination.	NE	NE	NE	Oman 98 strain; 100 TCID_50_	100% protected	[159]
	Hoti strain; Gn and Gc	BALB/c mice	Administered (SC) at dose 10^6^ particles at weeks 0, 2, and 5.	Yes	Yes	NE	NE	NE	[161]
	SPU 187/90 strain; NP	NIH-III Heterozygous mice strain	Administered (IM) at dose 100 μg of NP on days 0, 21, and 42. Administered (IM) at dose 50 μg of NP + 50 μg Poly (IC) on days 0, 21, and 42.	Yes	NE	Yes	NE	NE	[163]
	Hoti strain; NP and GPC	C57BL/6 mice with transiently immune suppressed by mAb-5A3	Administered (IM) at dose 2.5 μg of NP, 2.5 μ GPC, and 5 μg of NP + GPC on days 0 and 28.	Yes	Poor neutralization activity	Yes	UG3010; 100TCID_50_	100% protected for NP and NP + GPC, 37.5% protected for GPC	[164]
mRNA vaccines	Ank-2 strain; NP	C57BL/6 mice for immunogenicity and IFNAR^−/−^ mice for challenge studies	Administered (IM) at dose 25 μg of NP at weeks 0 and 2.	Yes	No	Yes	Ank-2 strain; 100LD_50_	Double dose immunized group showed 100% protection	[167]
	IbAr10200 strain; NP, Gn and Gc	IFNAR^−/−^ mice	Administered (ID) at doses 10 μg of NP, 10 μg of Gn, 10 μg of Gc, and 20 μg of NP + Gn + Gc at weeks 0 and 3.	Yes	Yes	Yes	IbAr10200 strain; 400FFU	100% protected	[168]
Viral vector-based vaccines	IbAr10200 strain; GPC expressed in modified vaccinia virus Ankara	IFNAR^−/−^ mice	Administered (IM) at dose 10^7^ PFU at weeks 0 and 2.	Yes	NE	Yes	IbAr10200 strain; 200TCID_50_	100% protected	[171]
	IbAr10200 strain; NP expressed in modified vaccinia virus Ankara	IFNAR^−/−^ mice	Administered (IM) at dose 10^7^ PFU by IM two times at weeks 0 and 2.	Yes	NE%	Yes	IbAr10200 strain; 200TCID_50_	Not protected	[172]
	IbAr10200 strain; NP expressed in Adenovirus type 5	IFNAR^−/−^ mice	Administered (IM) 1.25 × 10^7^ PFU for first dose at day 0. Administered (IN) 1 × 10^8^ PFU for second dose at day 28.	Yes	NE	NE	IbAr10200 strain; 1000LD_50_	78% protected	[173]
	Ank-2 strain; NP expressed in Bovine Herpesvirus Type 4 (BoHV-4)	BALB/c mice for serological assay and IFNAR^−/−^ mice for challenge studies	Administered (IP) at dose 100 TCID_50_ at weeks 0 and 2.	Yes	No	Yes	Ank-2 strain 100LD_50_	100% protected	[174]
	IbAr10200 strain; GPC expressed in ChAdOx2 (Chimpanzee Adenovirus)	BALB/c mice for immunogenicity and IFNAR^−/−^ mice for challenge studies	Administered (IM) at dose 5 × 10^7^ infectious unit (IU) on days 0 and 14.	Yes	Yes	Yes	IbAr10200 strain; 200FFU	100% protected	[175]
	IbAr10200 strain; GPC was expressed in Vesicular Stomatitis Virus expression system	STAT1 knock out mice	Administered (IP) at dose 10^7^ PFU on days 0 and 21.	Yes	Yes	NE	Turkey2004 strain; 50PFU	100% protected	[176]
	The CCHFV Nigeria IbAr10200 strain; CCHFV GPC was expressed by modified vaccinia virus ankara	IFNAR^−/−^ mice	10^7^ plaque-forming units were immunized (IM) two times at two week intervals.	Passively transferred the IgG and check the survival rate	NE	CD3+ T cells passively transferred and check the survival rate	IbAr10200; 200TCID_50_	Conferred 100% protection	[177]

NP—Nucleoprotein; GPC—Glycoprotein Complex; NE—Not evaluated; IM—Intramuscular; SC—Subcutaneous; ID—Intradermal; IP—Intraperitoneal; IN—Intranasal; PFU—Plaque Forming Unit; TCID_50_—Tissue Culture Infectious Dose 50; FFU—Fluorescence Focus Unit; LD_50_—Lethal dose 50; CD24—Cluster of differentiation 24; RVFV—Rift Valley Fiver Virus; HTNV—Hantaan Virus; TBEV—Tick-Borne Encephalitis Virus.

## 6. Conclusions

CCHF is a severe viral infection that poses a significant threat to public health. The high fatality rate and the absence of a specific treatment or vaccine make it crucial to understand the pathogenesis and immunology of CCHF to develop effective countermeasures. The use of animal models, including interferon-deficient mice and the Cynomolgus macaque, has significantly advanced the study of the disease’s mechanisms and potential treatments. Preclinical vaccine studies for various vaccine platforms have shown promising results. However, the genetic variability of CCHFV makes it challenging to develop a vaccine that can provide broad protection against all strains of the virus. Additionally, most CCHF vaccine studies have primarily employed the prototype IbAr10200 CCHFV strain, isolated in ticks, with uncertain human virulence. To address these challenges, heterologous challenge studies are required to develop more reliable vaccines that can provide broad protection against different strains of the CCHFV. Although preclinical studies show potential, these vaccines have not yet been tested in humans, and it remains to be determined if these findings can be successfully applied in human clinical trials.

## Figures and Tables

**Figure 1 diagnostics-13-02708-f001:**
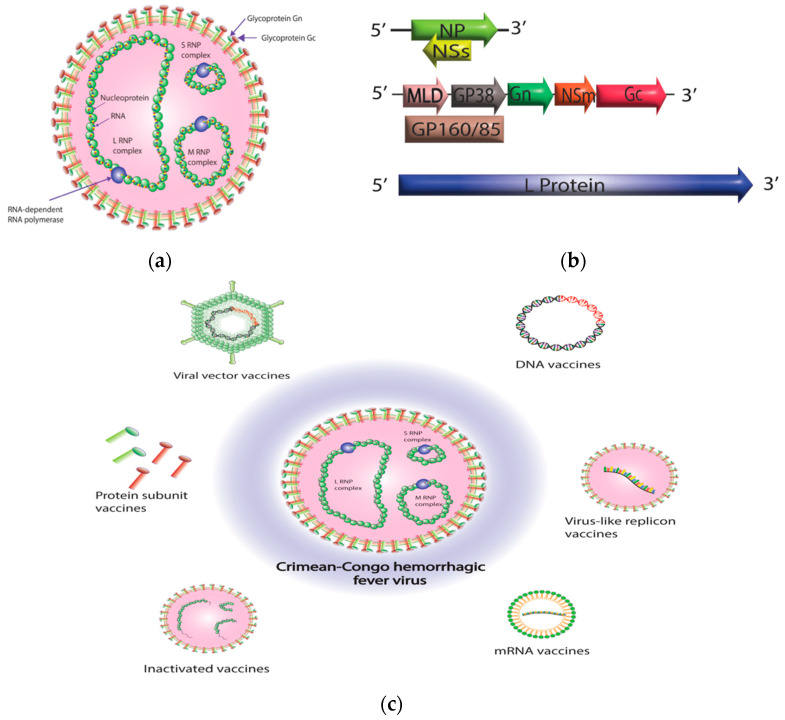
Crimean–Congo hemorrhagic fever virus (CCHFV) virion structure and CCHFV vaccine platforms. (**a**) CCHFV virion contains three single-stranded RNA segments with a negative-sense orientation. Nucleoprotein (NP) and RNA-dependent RNA polymerase (RdRp; L protein) protect the RNA by enclosing the RNA segments and forming ribonucleoprotein complexes (RNPs). Once the ribonucleoprotein (RNP) complexes are formed, they are surrounded by a protective envelope that originates from the membrane of the host cell. This envelope is coated with specialized glycoproteins known as Gn and Gc. (**b**) CCHFV consists of three genomic segments—small (S), medium (M), and large (L). The S segment is responsible for encoding the NP within one open reading frame, while the small non-structural protein (NSs) is encoded in an opposite-sense open reading frame. The M segment is quite intricate, as it encodes a glycoprotein precursor (GPC) that undergoes processing by host proteases. This processing results in the production of a GP160/85 domain, which is then further processed into a mucin-like domain (MLD) and GP38. Additionally, the M segment encodes the Gn and Gc glycoproteins, as well as the medium non-structural protein (NSm). The L segment of CCHFV, which is distinctively larger than other bunyaviruses, encodes for the viral RNA-dependent RNA polymerase (RdRP) and an ovarian tumor-like protease (OTU) at its N terminus. (**c**) A diagrammatic representation of different CCHFV vaccine platforms. The diagram was created with Adobe Illustrator.

## Data Availability

Not applicable.

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
