# Peer review of "Crimean–Congo Hemorrhagic Fever Virus: Progress in Vaccine Development"

_diagnostics, 2023, doi:10.3390/diagnostics13162708_

Round 1

Reviewer 1 Report

This paper is a comprehensive review of preclinical data on CCHFV vaccine. The paper has extended its coverage to mRNA vaccines and covers all platforms.

The main deficit on this paper is there is no road mapped for future work. These data are only valuable if you can prioritize platform of choice and discuss solutions to barriers. 

After reading this paper, I still have this question "what is the next step?" "Who and what product in which way should be put forward to move to phase 1 clinical trial?" As a funder or clinical trialist, I want to know which product is ready to go to pipeline and this paper does not address this question. 

This is the most challenging part of the paper and should be added to the paper for justifying this work. Otherwise, the paper is not adding much to the literature. We need some brainstorming and data analysis with a good plan on the table. 

Thanks 

I see this paper in good English writing skills. 

Author Response

Dear Reviewer,

Thank you for your feedback on our paper. We appreciate your comments.

Thank you for your continued attention to our paper. We appreciate your feedback and would like to provide further clarification on the points you raised.

We have already mentioned on the challenges associated with CCHFV vaccine development, including the genetic variability of the virus and the limitations of using the prototype IbAr 10200 CCHFV strain isolated in ticks. We indicated the need for heterologous challenge studies to develop vaccines that can provide broad protection against different strains of CCHFV.

Regarding the roadmap for future work, we have had a conclusion section that outlines potential directions for further research. We discuss the importance of conducting additional studies to evaluate the efficacy and safety of the vaccine candidates identified in preclinical studies. We also emphasize the need for human clinical trials to determine if these findings can be successfully translated into effective vaccines for CCHF.

While we have not specifically identified a single product ready for phase 1 clinical trials, we have provided an overview of the various vaccine platforms that have shown promising results in preclinical studies. The selection of a specific product for clinical trials involves careful consideration of factors such as safety, immunogenicity, scalability, and regulatory requirements. These decisions are typically made by research teams, funders, and regulatory bodies based on the available preclinical data and the overall landscape of vaccine development for CCHF.

Thank you for your valuable feedback.

Sincerely,

Reviewer 2 Report

Good review

Good

Author Response

Thank you for reading and evaluating the review, second reviewer.

Reviewer 3 Report

The manuscript titled "Crimean-Congo Hemorrhagic Fever Virus: Progress in Vaccine Development " provides a comprehensive overview and in-depth review of CCHFV and its vaccine development. It covering not only CCHFV epidemiology, transmission, geographic distribution, pathogenesis, immune response, but also the potential vaccine targets, animal models and the challenges in developing a vaccine that can provide broad protection against the genetically diverse CCHFV strains. The manuscript holds great significance for the field of CCHFV vaccine research. The paper is well-written and structured, with clear headings and subheadings that enable readers to grasp complex information with ease.

Author Response

I would like to express my sincere gratitude for your positive feedback on our manuscript titled "Crimean-Congo Hemorrhagic Fever Virus: Progress in Vaccine Development." I greatly appreciate your kind words and the time you took to provide a thorough review.